# Association of Psychosocial Factors on COVID-19 Testing among YWCA Service Recipients

**DOI:** 10.3390/ijerph20021297

**Published:** 2023-01-11

**Authors:** Miaya Blasingame, Veronica Mallett, Mekeila Cook, Wansoo Im, Derek Wilus, Robin Kimbrough, Gini Ikwuezunma, Ekemini Orok, Breia Reed, Victoria Akanbi, Aurdie Amoo-Asante, Maureen Sanderson

**Affiliations:** 1Vanderbilt University School of Medicine, Nashville, TN 37232, USA; 2Center for Women’s Health Research, Meharry Medical College, Nashville, TN 37208, USA

**Keywords:** psychosocial factors, COVID-19 testing, YWCA

## Abstract

The purpose of this study was to examine how psychosocial factors affect receipt of COVID-19 testing among Black and Hispanic women. In this cross-sectional study of Black and Hispanic women who received services from the YWCAs in Atlanta, El Paso, Nashville, and Tucson between 2019 and 2021 (*n* = 662), we used Patient-Reported Outcomes Measurement Information Systems (PROMIS) item bank 1.0 short forms to examine the impact of psychosocial factors (i.e., depression, anxiety, social isolation, instrumental support, emotional support, and companionship) on COVID-19 testing. Multivariable logistic regression models were used to estimate odds ratios and 95% confidence intervals for receipt of a COVID-19 test associated with psychosocial factors while adjusting for confounders. There was little effect of moderate/severe depressions or anxiety on receipt of COVID-19 testing. Black (odds ratio [OR] 0.58, 95% confidence interval [CI] 0.26–1.29) and Hispanic (OR 0.61, 95% CI 0.38–0.96) women with high levels of emotional support were less likely to receive the COVID-19 test. While high levels of instrumental support was associated with less likely receipt of the COVID-19 test among Black women (OR 0.75, 95% CI 0.34–1.66), it was associated with more likely receipt among Hispanic women (OR 1.19, 95% CI 0.74–1.92). Our findings suggest that certain psychosocial factors influence one’s decision to get a COVID-19 test which can be useful in encouraging preventive healthcare such as screening and vaccination.

## 1. Introduction

COVID-19 testing rates are lower among racial/ethnic minorities and low-income communities [1,2,3], and negative psychosocial factors may contribute to this disparity. Blacks and Hispanics, two groups that have been subjected to a history of racism and inequality in the United States, disproportionately experience stressors that lead to the development of negative psychosocial factors. Negative psychosocial factors, such as depression, anxiety, and social isolation, are associated with adverse health outcomes [4,5,6], while positive psychosocial factors, including social support and companionship, are known to benefit health [7,8,9]. Although Blacks and Hispanics were already subjected to stressors pre-COVID-19, the recent pandemic has resulted in an increase in individuals experiencing stressors and negative psychosocial factors [10,11,12], while also differentially impacting communities of color, with Black and Hispanic communities experiencing higher rates of COVID-19 exposure, infection, and mortality [13,14,15]. A recent study found that exposure to the psychosocial effects of the COVID-19 pandemic put adults, specifically women and Blacks, at high risk for depression and anxiety [16]. Another study reported racial/ethnic and SES disparities in physical and mental health status, including serious depression, during the pandemic [17]. Furthermore, psychosocial factors influence health behavior, with individuals suffering from negative psychosocial factors being less likely to engage in positive health behaviors [4,18]. Understanding the impact of both negative and positive psychosocial factors on COVID-19 testing could lead to increased testing among racial/ethnic minorities and a reduction in exposure and mortality.

While a few studies have focused on how structural racism drives COVID-19-related disparities for racial/ethnic minorities [15,19], there is a dearth of data available on the impact of psychosocial factors on COVID-19 testing, leaving the field blind to associations between the two. For example, many studies have found that structural factors like having fewer testing sites [20], residing in low-income neighborhoods [2,21], and discrimination [22] led to decreased testing among racial/ethnic minorities, but it would also be beneficial to understand if other barriers or facilitators, like psychosocial factors, are influencing receipt of COVID-19 testing as well. Disregarding the effect of psychosocial factors on testing can have serious implications for current and future testing initiatives and screening efforts.

Due to structural racism, Blacks and Hispanics often experience worse health outcomes compared to their White counterparts, and these same factors may increase vulnerability to COVID-19 [14,23,24]. Reports indicate that COVID-19 testing provides life-saving early detection, particularly for those with underlying medical conditions [25]. Early detection allows physicians an opportunity to employ multiple interventions to prevent or slow disease progression exacerbated by COVID-19. Since minority women tend to reside in multi-generational homes that may include non-family members, they have an increased risk of infecting other family and community members. Therefore, it is crucial for public health professionals to understand the factors contributing to minority women’s decisions to participate in COVID-19 testing. The purpose of this study is to examine how psychosocial factors, specifically depression, anxiety, social isolation, instrumental social support, emotional social support, and companionship, affect receipt of COVID-19 testing among Black and Hispanic women. Failing to understand how psychosocial factors influence COVID-19 testing is a serious epidemiologic oversight and allowing this lack of data to persist leaves public health professionals ignorant of additional barriers that may hinder COVID-19 and other infectious disease testing efforts.

## 2. Materials and Methods

### 2.1. Participants

The Towards Ending Societal Barriers to COVID-19 Testing in the United States (TEST-US Study), conducted from 1 February 2021 to 31 January 2022, examined factors associated with COVID-19 testing and vaccine uptake in minority women and their families. Participants were Black and Hispanic women who received services from the Young Women’s Christian Associations (YWCAs) in Atlanta, El Paso, Nashville, and Tucson between 1 February 2019 and 31 January 2021. In this community-engaged research project, Meharry Medical College’s Center for Women’s Health Research (CWHR) partnered with the YWCAs and a community advisory board (CAB) to design a mixed-methods study. Survey items were finalized with the assistance of our CAB and focus groups. The YWCA is the nation’s largest women’s organization serving over two million women and their families and is dedicated to women’s empowerment and the elimination of racism. YWCA provides housing for victims of domestic violence and their families, job training, education, and health services in over 200 locations across the United States [26]. Meharry Medical College, one of the nation’s oldest and most prestigious Historically Black Colleges and Universities (HBCU) [27], has a long history of serving the underserved and underrepresented populations of Tennessee, primarily Blacks and other minorities. The CWHR is devoted exclusively to understanding why women of color are at greater risk of certain diseases and how biology, race/ethnicity, and economics contribute to women’s health disparities. The TEST-US Study was reviewed and approved by the institutional review board of Meharry Medical College.

### 2.2. Methods

To be eligible for the study, women had to self-identify as Black or Hispanic, to be age 18 or older, to have received YWCA services within the two years prior to survey completion, and to have provided the YWCA with contact information. YWCA staff contacted eligible women by phone, text, email, or mail with a request to complete a phone, online, or self-administered paper survey. We were unable to identify the total number of potential participants since contact information may have changed over time. All participants read the information sheet and provided implied consent by completing the survey. Survey topics included: demographics, COVID-19-related factors such as infection/testing/vaccine/treatment, living situation, interpersonal violence, health and healthcare, smoking and alcohol use, emotional health, and support system. After exclusions for answering fewer than 5 questions (*n* = 5) and missing COVID-19 testing (*n* = 1), data were available for 662 women who completed surveys, primarily online, from 24 June 2021 to 24 December 2021. In addition to individual-level measures, we collected neighborhood-level measures corresponding to the residence of the YWCA service recipients.

Women were asked if they had been tested for COVID-19, if they were unable to get a COVID-19 test, and, if applicable, the reasons they were unable to get a COVID-19 test. The Patient-Reported Outcomes Measurement Information Systems (PROMIS) item bank 1.0 short forms 4a (available from https://www.healthmeasures.net, accessed on 15 August 2022) were utilized to gather information on psychosocial factors of depression, anxiety, social isolation, instrumental support (e.g., “Do you have someone who can help you if you are confirmed to bed?”), emotional support, and companionship. The scales pertained to the past seven days and asked the frequency with which each feeling occurred, ranging from never (1) to always (5). All scales had very high internal consistency with Cronbach’s alphas ranging from 0.93 for anxiety to 0.96 for instrumental and emotional support. Raw scale scores ranging from 4 to 20 were submitted to the HealthMeasures Scoring Service (Assessment Center Scoring Service, n.d.) to compute T-scores. For depression, anxiety, and social isolation, T-scores of 40 to <55 were classified as within normal limits (WNL), 55 to <60 as mild, and 60 to 82 as moderate/severe. For emotional support, instrumental support and companionship, T-scores of 25 to <40 were classified as very low/low, 40 to <60 as average, and 60 to 75 as high/very high. Respondents with more than one missing scale item were excluded from analyses, yielding totals for subsequent analyses ranging from 609 for emotional support to 617 for depression.

### 2.3. Data Analysis

Frequency distributions of demographic and psychosocial factors by COVID-19 testing and race/ethnicity were examined using chi-square statistics. Multivariable logistic regression models were used to estimate odds ratios (OR) and 95% confidence intervals (CI) for receiving a COVID-19 test associated with psychosocial factors while adjusting for confounding factors. Potential confounders of these associations were survey completion date, age, language, educational attainment, household income, marital status, current employment, household living situation, and general health status. We stratified by race/ethnicity *a priori* since we did not have sufficient statistical power to examine effect modification. All statistical analyses were performed using SAS software version 9.4 (SAS Institute Inc., Cary, NC, USA).

## 3. Results

Of the 662 women who completed surveys, 199 (30.1%) were Black and 463 (69.9%) were Hispanic. The majority of surveys were completed in June and July 2021 among women aged 40 years and older.

Table 1 presents demographic and psychosocial factors of participants by COVID-19 testing. Although not significantly different, the earlier the survey was completed, the less likely women had been tested for COVID-19. The only factors that differed significantly by receipt of the COVID-19 test were marital status (*p* = 0.05) and household living situation (*p* = 0.04). Subsequent analyses were adjusted for these three variables.

Table 2 presents demographic and psychosocial factors of participants by race/ethnicity. The only factor that did not differ significantly by race/ethnicity was general health status (*p* = 0.91).

Table 3 presents the association between psychosocial factors and COVID-19 testing stratified by race/ethnicity. There was little effect of moderate/severe depression (Black OR 1.03, 95% CI 0.44–2.38, Hispanic OR 0.95, 95% CI 0.56–1.60) or anxiety (Black OR 1.09, 95% CI 0.47–2.51, Hispanic OR 0.99, 95% CI 0.62–1.59) on receipt of COVID-19 testing. Black (OR 0.58, 95% CI 0.26–1.29) and Hispanic (OR 0.61, 95% CI 0.38–0.96) women with high levels of emotional support were less likely to receive the COVID-19 test. While high levels of instrumental support was associated with less likely receipt of the COVID-19 test among Black women (OR 0.75, 95% CI 0.34–1.66), it was associated with more likely receipt among Hispanic women (OR 1.19, 95% CI 0.74–1.92).

## 4. Discussion

### 4.1. Summary of the Findings

In this study of a diverse group of Black and Hispanic women, we found little effect of negative psychosocial factors on receipt of COVID-19 testing. Neither depression nor anxiety had a significant impact on whether Black or Hispanic women received a COVID-19 test. These results appear to be counterintuitive, but the seemingly insignificant influence of negative psychosocial factors on COVID-19 testing could be due to the impact of resilience and social support. While the COVID-19 pandemic undoubtedly resulted in increased levels of stress, depression, and anxiety for many individuals, it is not unreasonable for people to exhibit resilience during crises, particularly among racial and ethnic minority groups. Some research suggests that racial and ethnic minorities have displayed high levels of resilience despite experiencing inequality and discrimination [28,29]. One key component of resilience is social support. The benefits of social support on mental health and the development of resilience are well documented in the literature [30,31]. Furthermore, our findings support prior research on the importance of social support.

Black women with high levels of emotional and instrumental support were less likely to receive a COVID-19 test, while Hispanic women with high levels of instrumental support were more likely to receive a COVID-19 test. For Black respondents, one possible explanation for these results is that these women are receiving support from members of their own racial community, where the long-lasting effects of medical racism contribute to feelings of mistrust toward medical professionals and programs, hindering their willingness to get tested or encourage others to get tested [21,32]. Additionally, the spread of misinformation about COVID-19 and testing through social media networks may contribute to women’s decision to be tested [21,33]. Since support networks tend to be trusted sources of information for individuals, the women in our study may have chosen to trust their support networks over validated medical information, and therefore decided to not get a COVID-19 test. As for Hispanic women, instrumental support may have facilitated their receipt of a COVID-19 test by alleviating some of the barriers these women faced when trying to get tested. For example, those providing instrumental support to the women in our study may have been able to facilitate COVID-19 testing by addressing identified barriers like transportation to testing sites, childcare while at testing sites, help with caregiving if a positive test is received, or financial assistance in case of job loss [32,34].

### 4.2. Limitations

There are a few limitations to this study that must be addressed. First, the cross-sectional nature of this study limits our ability to draw causal inference or identify temporal relationships. However, this project was a part of a novel partnership between Meharry and the YWCA. This partnership with the YWCA allows for a diverse cross-section of women and their families to participate in a future longitudinal cohort study, which will be beneficial for drawing causal inferences. Second, quantitative surveys are potentially limited by selection bias and information bias. We utilized rigorous tracking and follow-up procedures to attain the projected response rate of 60% and the survey was designed to minimize information bias; however, the survey did contain questions on racial/ethnic discrimination which may be sensitive for respondents to answer and therefore prone to misclassification. Another limitation is the availability of COVID-19 tests during the study period. Early in the pandemic, testing kits were reserved for specific populations, for example, people with underlying medical conditions. Such restrictions may have affected the availability of testing for women who did not meet the criteria for receiving a test, which could have influenced their decision to be tested. We controlled for survey completion date as a proxy for testing eligibility. The percentage of women missing data on psychosocial factors differed by testing status among Black women (no 12%; yes 3%), but was comparable among Hispanic women (no 6%; yes 8%). This may have resulted in an overestimate of effects among Black women, but not among Hispanic women. Lastly, the study recruitment from YWCAs may limit generalizability of research findings since each YWCA focused on providing different services (e.g., Atlanta provided mammograms to older women while Nashville provided a domestic violence shelter).

### 4.3. Strengths

Despite its limitations, this study has several strengths. First, the community-engaged nature of this study—particularly the multidisciplinary team of investigators who are committed to achieving health equity and our established partnerships with community partners and the strong leadership on our CAB—enhanced our ability to successfully implement community-engaged research and help ensure that our survey was culturally appropriate and relevant to the community. Second, the study’s focus on a target population of fairly low income Black and Hispanic women provides much needed data on an understudied group that is at increased risk of COVID-19 incidence and mortality. Third, the PROMIS measures used in this study to collect information on psychosocial factors are well-developed and validated for use in the general population, and also have been found to be valid and reliable among multiple races/ethnicities and ages [35].

### 4.4. Practical Implications

The TEST-US study is innovative in that there are no existing community engaged research projects addressing how psychosocial factors impact COVID-19 testing among racial/ethnic minority women living in resource-restricted communities. Previous studies primarily address the incidence and mortality rates of COVID-19 among minority populations and their access to COVID-19 testing, without taking psychosocial factors into account. Racial/ethnic minorities are more likely to experience factors associated with structural racism, which increase their risk of experiencing negative psychosocial factors. These factors may have been exacerbated by the pandemic and instead of attributing minorities’ reluctance to COVID-19 testing as poor health behavior, public health professionals must understand the structural factors that might result in low testing among this group.

Future studies examining COVID-19 testing decisions should include additional relevant psychosocial factors like bereavement and job insecurity—especially since racial/ethnic minorities have the highest mortality rates from COVID-19 and have less control over their jobs, specifically in regard to being able to miss work due to COVID-19 exposure or practice safety precautions like social distancing while on the job or being able to work from home. Future studies may also consider the cultural implications of specific coping strategies on COVID-19 testing behaviors. Future studies may also benefit from examining specific sources of social support, such as spousal, friend, coworker, or family social support, to see if a particular source of social support is more beneficial in encouraging racial/ethnic minorities to receive COVID-19 testing. It would also be beneficial for future studies to examine the mediating effect of different types of social support on the impact of psychosocial factors on likeliness to get tested for COVID-19.

Ultimately, the results from this study will be used to expand geographic representation from YWCA sites nationally for a future longitudinal study. These results will help center the impact of psychosocial factors on minority health, especially since the pandemic has resulted in an increase in people experiencing negative psychosocial factors. Our intention is to develop interventions and inform policy for minimizing distress and death caused by the pandemic.

## 5. Conclusions

Our findings suggest that certain psychosocial factors influence one’s decision to get a COVID-19 test, which can be useful in encouraging preventive healthcare such as screening and vaccination. The findings from this study could be used to determine the best way to encourage adoption of health behaviors focused on preventive care. Specifically, the impact of emotional and instrumental support on testing decisions could be useful for testing efforts for future pandemics, endemic viruses like the flu, and screening efforts for cancers and other diseases. Testing is a means of screening for many diseases, but compared to their White counterparts, racial/ethnic minorities are less likely to receive preventive screenings and more likely to be diagnosed at later stages of diseases where screening is available [36]. Understanding how psychosocial factors and social support contribute to women’s low COVID-19 screening rates is crucial in order to implement effective testing initiatives. Furthermore, many of the barriers and influences that affect one’s decision to get a COVID-19 test may also contribute to hesitancy surrounding receipt of the COVID-19 vaccine. Vaccine hesitancy has been an issue before COVID-19, for example, vaccine hesitancy surrounding the HPV vaccine [37], and will continue to be an issue unless vaccine promotion efforts take personal influences like psychosocial factors and social support into account, in addition to structural factors.

## Figures and Tables

**Table 1 ijerph-20-01297-t001:** Demographic and psychosocial factors by COVID-19 testing, TEST-US Study, 2021.

	No (*n* = 183)		Yes (*n* = 479)		
Factor	Number	%	Number	%	*p*-Value
Race/ethnicity					
Black	52	28.4	147	30.7	0.57
Hispanic	131	71.6	332	69.3	
Completion date					
June	63	34.4	129	26.9	0.28
July	49	26.8	145	30.3	
August	22	12.0	80	16.7	
September	29	15.9	77	16.1	
October-December	20	10.9	48	10.0	
Age					
18–29 years	21	12.2	53	11.7	0.41
30–39 years	44	25.6	117	25.8	
40–49 years	47	27.3	152	33.5	
50–87 years	60	34.9	132	29.0	
Missing	11		25		
Language					
English	122	66.7	338	70.6	0.33
Spanish	61	33.3	141	29.4	
Education					
≤High school	60	33.0	157	32.8	0.36
Some college/vocational school	71	39.0	163	34.0	
College graduate+	51	28.0	159	33.2	
Missing	1		0		
Income					
<$15,000	63	34.8	128	27.3	0.27
$15,000–24,999	33	18.2	104	22.2	
$25,000–49,999	42	23.2	123	26.2	
$50,000+	43	23.8	114	24.3	
Missing	2		10		
Marital status					
Single	39	21.4	147	30.8	0.05
Married	84	46.2	200	41.9	
Separated, widowed, divorced	59	32.4	130	27.3	
Missing	1		2		
Current employment					
Work full time	68	38.2	204	44.8	0.27
Work part time/unemployed	76	42.7	181	39.8	
Retired/homemaker/disability/student	34	19.1	70	15.4	
Missing	5		24		
Household living situation					
Lives alone	31	17.9	53	11.8	0.04
Lives with others	142	82.1	398	88.2	
Missing	10		28		
General health status					
Fair/poor	35	20.8	81	18.0	0.41
Good	54	32.2	170	37.7	
Very good/excellent	79	47.0	200	44.3	
Missing	15		28		
Depression					
Within normal limits	77	45.6	220	49.2	0.54
Mild	44	26.0	98	21.9	
Moderate/severe	48	28.4	129	28.9	
Missing	14		32		
Anxiety					
Within normal limits	78	45.9	203	45.9	0.80
Mild	28	16.5	82	18.6	
Moderate/severe	64	37.7	157	35.5	
Missing	13		37		
Social isolation					
Within normal limits	132	78.1	340	77.3	0.75
Mild	17	10.1	53	12.0	
Moderate/severe	20	11.8	47	10.7	
Missing	14		39		
Emotional support					
Very low/low	23	13.5	62	14.2	0.12
Average	73	43.0	224	51.1	
High	74	43.5	152	34.7	
Missing	13		41		
Instrumental support					
Very low/low	26	15.2	91	20.7	0.24
Average	91	53.2	207	47.2	
High	54	31.6	141	32.1	
Missing	12		40		
Companionship					
Very low/low	22	12.9	70	15.9	0.29
Average	95	55.5	257	58.4	
High/very high	54	31.6	113	25.7	
Missing	12		39		

**Table 2 ijerph-20-01297-t002:** Demographic and psychosocial factors by race/ethnicity, TEST-US Study, 2021.

	Black (*n* = 199)		Hispanic (*n* = 463)		
Factor	Number	%	Number	%	*p*-Value
Completion date					
June	14	7.0	178	38.4	<0.0001
July	46	23.1	148	32.0	
August	35	17.6	67	14.5	
September	63	31.7	43	9.3	
October-December	41	20.6	27	5.8	
Age					
18–29 years	23	12.6	51	11.5	<0.0001
30–39 years	25	13.6	136	30.7	
40–49 years	40	21.9	159	35.9	
50–87 years	95	51.9	97	21.9	
Missing	16		20		
Language					
English	198	99.5	262	56.6	<0.0001
Spanish	1	0.5	201	43.4	
Education					
≤High school	74	37.2	143	31.0	0.001
Some college/vocational school	82	41.2	152	32.9	
College graduate+	43	21.6	167	36.1	
Missing	0		1		
Income					
<$15,000	82	42.3	109	23.9	<0.0001
$15,000–24,999	39	20.1	98	21.5	
$25,000–49,999	35	18.0	130	28.5	
$50,000+	38	19.6	119	26.1	
Missing	5		7		
Marital status					
Single	100	50.8	86	18.6	<0.0001
Married	31	15.7	253	54.8	
Separated, widowed, divorced	66	33.5	123	26.6	
Missing	2		1		
Current employment					
Work full time	65	34.4	207	46.6	0.0002
Work part time/unemployed	100	52.9	157	35.4	
Retired/homemaker/disability/student	24	12.7	80	18.0	
Missing	10		19		
Household living situation					
Lives alone	44	23.8	40	9.1	<0.0001
Lives with others	141	76.2	399	90.9	
Missing	14		24		
General health status					
Fair/poor	37	19.8	79	18.3	0.91
Good	67	35.8	157	36.3	
Very good/excellent	83	44.4	196	45.4	
Missing	12		31		
Depression					
Within normal limits	107	56.9	190	44.4	0.008
Mild	31	16.5	111	25.9	
Moderate/severe	50	26.6	127	29.7	
Missing	11		35		
Anxiety					
Within normal limits	90	48.1	191	44.9	0.04
Mild	42	22.5	68	16.0	
Moderate/severe	55	29.4	166	39.1	
Missing	12		38		
Social isolation					
Within normal limits	142	75.9	330	78.2	0.03
Mild	16	8.6	54	12.8	
Moderate/severe	29	15.5	38	9.0	
Missing	12		41		
Emotional support					
Very low/low	42	22.8	43	10.1	<0.0001
Average	88	47.8	209	49.3	
High	54	29.4	172	40.6	
Missing	15		39		
Instrumental support					
Very low/low	47	25.3	70	16.5	0.04
Average	86	46.2	212	50.0	
High	53	28.5	142	33.5	
Missing	13		39		
Companionship					
Very low/low	49	26.5	43	10.1	<0.0001
Average	110	59.5	242	56.8	
High/very high	26	14.0	141	33.1	
Missing	14		37		

**Table 3 ijerph-20-01297-t003:** Association between psychosocial factors and COVID-19 testing stratified by race/ethnicity, TEST-US Study, 2021.

**Black**					
	**No**		**Yes**		
**Factor**	**Number**	**%**	**Number**	**%**	**OR * (95% CI)**
Depression					
Within normal limits	27	58.7	80	56.3	1.00 (referent)
Mild	7	15.2	24	16.9	1.16 (0.43–3.11)
Moderate/severe	12	26.1	38	26.8	1.03 (0.44–2.38)
Missing	6		5		
Anxiety					
Within normal limits	23	50.0	67	47.5	1.00 (referent)
Mild	9	19.6	33	23.4	1.17 (0.47–2.89)
Moderate/severe	14	30.4	41	29.1	1.09 (0.47–2.51)
Missing	6		6		
Social isolation					
Within normal limits	35	76.1	107	75.9	1.00 (referent)
Mild	3	6.5	13	9.2	1.84 (0.37–9.16)
Moderate/severe	8	17.4	21	14.9	0.84 (0.32–2.23)
Missing	6		6		
Emotional support					
Very low/low	11	23.9	31	22.5	0.70 (0.28–1.79)
Average	19	41.3	69	50.0	1.00 (referent)
High	16	34.8	38	27.5	0.58 (0.26–1.29)
Missing	6		9		
Instrumental support					
Very low/low	10	21.7	37	26.4	1,14 (0.46–2.87)
Average	21	45.7	65	46.4	1.00 (referent)
High	15	32.6	38	27.2	0.75 (0.34–1.66)
Missing	6		7		
Companionship					
Very low/low	11	23.9	38	27.4	1.22 (0.51–2.93)
Average	26	56.5	84	60.4	1.00 (referent)
High/very high	9	19.6	17	12.2	0.50 (0.19–1.30)
Missing	6		8		
**Hispanic**					
	**No**		**Yes**		
**Factor**	**Number**	**%**	**Number**	**%**	**OR * (95% CI)**
Depression					
Within normal limits	50	40.6	140	45.9	1.00 (referent)
Mild	37	30.1	74	24.3	0.69 (0.41–1.16)
Moderate/severe	36	29.3	91	29.8	0.95 (0.56–1.60)
Missing	8		27		
Anxiety					
Within normal limits	55	44.4	136	45.2	1.00 (referent)
Mild	19	15.3	49	16.3	1.07 (0.57–1.99)
Moderate/severe	50	40.3	116	38.5	0.99 (0.62–1.59)
Missing	7		31		
Social isolation					
Within normal limits	97	78.9	233	77.9	1.00 (referent)
Mild	14	11.3	40	13.4	1.32 (0.67–2.60)
Moderate/severe	12	9.8	26	8.7	0.89 (0.42–1.87)
Missing	8		33		
Emotional support					
Very low/low	12	9.7	31	10.3	0.89 (0.41–1.92)
Average	54	43.5	155	51.7	1.00 (referent)
High	58	46.8	114	38.0	0.61 (0.38–0.96)
Missing	7		32		
Instrumental support					
Very low/low	16	12.8	54	18.1	1.86 (0.96–3.59)
Average	70	56.0	142	47.5	1.00 (referent)
High	39	31.2	103	34.4	1.19 (0.74–1.92)
Missing	6		33		
Companionship					
Very low/low	11	8.8	32	10.6	1.23 (0.56–2.71)
Average	69	55.2	173	57.5	1.00 (referent)
High/very high	45	36.0	96	31.9	0.74 (0.47–1.18)
Missing	6		31		

* Odds ratio adjusted for survey completion date, marital status, and household living situation.

## Data Availability

Data available upon request.

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
