# Peer review of "Association of Psychosocial Factors on COVID-19 Testing among YWCA Service Recipients"

_ijerph, 2023, doi:10.3390/ijerph20021297_

Round 1

Reviewer 1 Report

The authors conducted a cross-sectional observational study to examine the associations between psychosocial factors and receipt of COVID-19 testing among Black and Hispanic women. By analyzing the data of 662 women, the authors showed that there were no significant associations between psychosocial factors (depression, anxiety, social isolation, emotional support, instrumental support, companionship) and receipt of COVID-19 testing.

There are some comments.

Comments:

1.      Materials and Methods (Line 102 on Page 3): “YWCA staff contacted eligible women by phone, email, -.” Please describe the eligibility criteria before the sentence. In addition, it is unclear how many women were screened for eligibility. It is also unclear how many women were eligible, how many women were ineligible, and how many eligible women agreed to participate in the study. A more detailed description is recommended.

2.      Materials and Methods (Line 108 on Page 3): “After exclusions for ineligibility and missing COVID-19 testing, data were available for 662 women who completed surveys, -.” It is unclear how many women were ineligible and how many women had missing COVID-19 testing data. In addition, it is unclear whether there were differences and sociodemographic or psychosocial status between the excluded and included women. A presentation of the demographic and psychosocial factors of the excluded individuals is suggested. A comparison of the demographic and psychosocial factors of the excluded individuals with those of the included individuals is also recommended.

3.      Materials and Methods (Lines 114-128 on Page 3): “The Patient-Reported Outcomes Measurement Information Systems (PROMIS) item bank 1.0 short forms 4a were utilized to gather information on -.” The authors described the methods of measuring the psychosocial factors, including the questionnaires, scoring procedures, and categorization. However, the references supporting these methods were lacking.

4.      Discussion: As shown in Table 3, there were missing data on each psychosocial factor in both Black and Hispanic women. The authors decided to take a complete-case-analysis approach. Would the data missingness have any effects on the analytic results? A discussion of this limitation is recommended.   

5.      Discussion: This study recruited participants from YWCAs. Would this affect the generalizability of the research findings? A discussion of this limitation is suggested.

6.      Conclusions: Please begin the conclusion with a summary of the findings.

7.      Abstract (Line 29 on Page 1): “Black and Hispanic women with high levels of emotional support were less likely to receive the COVID-19 test.” Please add the estimated odds ratio.

8.      Abstract (Line 31 on Page 1): “it was associated with more likely receipt among Hispanic women.” Please add the estimated odds ratio.

9.      Title: This is an observational study. Please avoid the term “impact.” An example of a more appropriate title would be “Association of psychosocial factors on COVID-19 testing among YWCA service recipients.”

Reviewer 2 Report

I congratulate the authors on their interesting research and believe that the manuscript can be recommended for publication after minor revisions:

1.      The authors formulate the purpose of the study as follows: “The purpose of this study is to examine how psychosocial factors, specifically depression, anxiety, social isolation, instrumental social support, emotional social support, and companionship affect receipt of COVID-19 testing among Black and Hispanic women”. In the theoretical review, the authors refer to studies of anxiety, social isolation, and social isolation, but do not write about depression in the context of the topic. I think it would be helpful to mention the results of comparative studies of depression.

2.      For the reader's convenience, I recommend dividing the "Materials and Methods" section into the traditional subsections "Participants," "Methods," and "Data Analysis". It also seems to me advisable to highlight in the "Discussion" section the subsection "Limitations" and add a subsection "Practical Implications". In the abstract the authors write the following: “Our findings suggest that certain psychosocial factors influence one’s decision to get a COVID-19 test which can be useful in encouraging preventive healthcare such as screening and vaccination”. This conclusion with its interpretations can either be described in more detail in a separate subsection, or it can be placed in the "Conclusion" section.
